

# Influence of mutation bias and hydrophobicity on the substitution rates and sequence entropies of protein evolution

María José Jiménez-Santos[1], Miguel Arenas[2] and Ugo Bastolla[1]

[1] Bioinformatics Unit, Center for Molecular Biology Severo Ochoa, CSIC-UAM, Madrid, Spain
[2] Department of Biochemistry, Genetics and Immunology, University of Vigo, Vigo, Spain

## ABSTRACT

The number of amino acids that occupy a given protein site during evolution reflects the selective constraints operating on the site. This evolutionary variability is strongly influenced by the structural properties of the site in the native structure, and it is quantified either through sequence entropy or through substitution rates. However, while the sequence entropy only depends on the equilibrium frequencies of the amino acids, the substitution rate also depends on the exchangeability matrix that describes mutations in the mathematical model of the substitution process. Here we apply two variants of a mathematical model of protein evolution with selection for protein stability, both against unfolding and against misfolding. Exploiting the approximation of independent sites, these models allow computing site-specific substitution processes that satisfy global constraints on folding stability. We find that site-specific substitution rates do not depend only on the selective constraints acting on the site, quantified through its sequence entropy. In fact, polar sites evolve faster than hydrophobic sites even for equal sequence entropy, as a consequence of the fact that polar amino acids are characterized by higher mutational exchangeability than hydrophobic ones. Accordingly, the model predicts that more polar proteins tend to evolve faster. Nevertheless, these results change if we compare proteins that evolve under different mutation biases, such as orthologous proteins in different bacterial genomes. In this case, the substitution rates are faster in genomes that evolve under mutational bias that favor hydrophobic amino acids by preferentially incorporating the nucleotide Thymine that is more frequent in hydrophobic codons. This appearingly contradictory result arises because buried sites occupied by hydrophobic amino acids are characterized by larger selective factors that largely amplify the substitution rate between hydrophobic amino acids, while the selective factors of exposed sites have a weaker effect. Thus, changes in the mutational bias produce deep effects on the biophysical properties of the protein (hydrophobicity) and on its evolutionary properties (sequence entropy and substitution rate) at the same time. The program Prot_evol that implements the two site-specific substitution processes is freely available at https://ub.cbm.uam.es/prot_fold_evol/prot_fold_evol_soft_main.php#Prot_Evol.

Corresponding author
Ugo Bastolla, ubastolla@cbm.csic.es

## INTRODUCTION

The evolutionary variability of an amino acid site in a protein family is an important indicator of the selective constraints that the site experiences. This variability is usually quantified either through the sequence entropy (e.g., *Goldstein & Pollock, 2017*) or through the substitution rate (e.g., *Grishin, Wolf & Koonin, 2000*). These two measures of evolutionary variability are considered to be essentially equivalent, see for instance the arguments presented in the seminal paper by *Halpern & Bruno (1998)*. Here we adopt a model of protein evolution with global selective constraints for the maintenance of the thermodynamic stability of the native state both against unfolding and against misfolding, and we show that these two measures of evolutionary variability are not in general equivalent since they are differently influenced by the mutational process, which in general favors exchanges between polar amino acids, so that for sites with equal sequence entropy the site-specific substitution rate tends to be higher at exposed sites occupied by polar amino acids. Because of the same reason, we find that substitution rates averaged across sites of the same protein are higher for more polar proteins. However, when we compare different mutational processes, we find the counterintuitive result that mutational processes that favor hydrophobic residues, such as those taking place in the genomes of AT rich intracellular bacteria, tend to favor higher substitution rates. This is a result that we argue is due to the differential constraints imposed by natural selection on buried and exposed sites.

The evolutionary variability of a protein site is strongly influenced by the structural properties of the site in the native state of the protein (*Echave, Spielman & Wilke, 2016*). In particular, the substitution rate changes dramatically between exposed and buried sites in such a way that buried sites tend to evolve more slowly than exposed sites. This is generally attributed to the fact that natural selection imposes stronger constraints on buried sites (*Franzosa & Xia, 2009*). It was later shown that the number of native inter-residue contacts formed by a protein site, which is negatively correlated with the solvent accessibility, is a stronger predictor of the substitution rate (*Yeh et al., 2014*).

Two different models rationalize why sites that form many contacts are subject to stronger selective constraints. The first kind of model, which we call stability-constrained fitness model, models the fitness as the fraction of protein found in the native state, which is a sigmoidal function of the folding free energy $\Delta G$, i.e., $f = 1/(1 + \exp(-\Delta G/kT))$ (see *Goldstein, 2011*; *Serohijos & Shakhnovich, 2014*; *Bastolla, Dehouck & Echave, 2017*). The second kind of model is the structurally-constrained model of protein evolution, which estimates how mutations affect the structure of the native state and computes the fitness from this predicted structural change (*Echave, 2008*). In the literature stability-constrained models are sometimes called structurally-constrained models, but we think that this wording is misleading, since the fitness function that they assume depends only on stability and not on structural changes. On the other hand, structurally-constrained models model the mutation as a perturbation that changes the wild-type as predicted through the Elastic Network Model (ENM *Tirion, 1996*) and linear response theory and assume that the stability does not change. Thus, stability-constrained models predict the effect of mutations

through the predicted stability change but neglect the effect of the corresponding structure change, and structure-constrained models adopt the complementary perspective. Of course mutations modify both the stability and the precise structure of the native state, but current models of fitness cannot compute both effects.

In a recent work, we have shown that stability-constrained models that take into account negative design for destabilizing misfolded conformations (*Berezovsky, Zeldovich & Shakhnovich, 2007*; *Noivirt-Brik, Horovitz & Unger, 2009*; *Minning, Porto & Bastolla, 2013*) predict that both the substitution rate and the entropy are maximal not at exposed sites with few contacts, as observed, but at sites where the number of contacts is intermediate, which can accomodate both hydrophobic and polar amino acids and are predicted to be extremaly tolerant to mutations (*Jimenez, Arenas & Bastolla, 2018*). On the other hand, when stability with respect to misfolding is neglected, stability-constrained models predict that the variability is maximal at exposed sites with few contacts (*Scherrer, Meyer & Wilke, 2012*; *Echave, Jackson & Wilke, 2015*), but these kinds of models overestimate both the tolerance to mutations and the average hydrophobicity at almost all positions (*Jimenez, Arenas & Bastolla, 2018*) and they score much worse than models that consider misfolding in likelihood calculations (*Arenas, Sanchez-Cobos & Bastolla, 2015*), so that models that consider misfolding have to be preferred. In contrast, structure-constrained models correctly predict that the variability is inversely related with the number of native contacts (*Huang et al., 2014*). These results support the view that the structural effect of mutations cannot be neglected, in particular at sites with intermediate numbers of contacts that are extremely tolerant to mutations under the point of view of the stability.

Here we adopt the stability-constrained mean-field (MF, *Arenas, Sanchez-Cobos & Bastolla, 2015*; *Bastolla et al., 2006*) and wild-type (WT, *Jimenez, Arenas & Bastolla, 2018*) models of protein evolution that we used in the above-mentioned study. These models assume that sites in the protein evolve independently in a site-specific manner, and determine their site-specific properties by imposing a global constraint on the thermodynamic stability of the known native state against both unfolding and misfolding. The MF model significantly improves the likelihood of inferred evolutionary events with respect to empirical models that do not take into account the structural properties of each site (*Arenas, Sanchez-Cobos & Bastolla, 2015*), and it improves the reconstruction of the stability properties of ancestral sequences (*Arenas et al., 2017*). The WT model shows even better performances on several data sets (M Arenas & U Bastolla, in preparation). Both models exploit the formal analogy between the Boltzmann distribution in statistical physics, in which the probability of each conformation depends on the energy changed of sign and on the inverse of the temperature, and the stationary distribution of a protein family in which the probability of each sequence depends on its fitness and on the effective population size (*Sella & Hirsh, 2005*; *Mustonen & Lässig, 2005*).

In the MF model, the effect on stability of amino-acid $a$ at site $i$ is predicted self-consistently against the MF distribution at all other sites, in the spirit of mean-field models in statistical mechanics. In turn, the WT model predicts the effect on stability and fitness of mutations of the wild-type sequence towards amino acid $a$ at site $i$. Thus, in theory the

WT model is more suited for short evolutionary divergences and the MF model is more suited for long evolutionary divergences (M Arenas & U Bastolla, in preparation).

After the site-specific amino-acid frequencies have been determined, the site-specific exchangeability matrices that allow constructing the full site-specific substitution process are computed applying the Halpern and Bruno formulas (*Halpern & Bruno, 1998*), which impose that the fixation probabilities agree with Kimura's formulas (*Kimura, 1962*). Both formulas are reproduced below for completeness.

Here we address the question whether the sequence entropy and the substitution rate are equivalent measures of the evolutionary variability of a position, as suggested by *Halpern & Bruno (1998)* who argue that these measures should be positively related in general. Nevertheless, we find that these two measures are not equivalent, since the sequence entropy is only influenced by the equilibrium distribution of amino acids while the substitution rate is also influenced by the mutation process that acts in evolution. In particular, for equal sequence entropy, sites that are preferentially occupied by amino acids with higher exchangeability have higher substitution rate, so that the substitution rate is not a monotonic function of the sequence entropy in general. We also found that polar amino acids are characterized by higher exchangeability so that, for equal sequence entropy, exposed sites occupied by polar amino acids tend to substitute faster. However, it is a bit counterintuitively, but expected on the basis of the present model, that if we simulate mutational processes that favor hydrophobic amino acids the substitution rate increases and the maximum across sites of the substitution rate moves towards more hydrophobic sites, so that which sites are substituted faster ultimately depends on the mutation bias.

## MATERIALS AND METHODS

### Stability constrained fitness model

Stability constrained models of protein evolution assume that the fitness of a protein with sequence $\mathbf{A}$ is proportional to the fraction of protein that is in the native state, which can be computed from the folding free energy as (*Goldstein, 2011*; *Serohijos & Shakhnovich, 2014*)

$$f(\mathbf{A}) = e^{-\Delta G(\mathbf{A})/kT} / \left(1 + e^{-\Delta G(\mathbf{A})/kT}\right). \tag{1}$$

The computation is performed assuming that the native contact matrix $C^{\mathrm{nat}}$ does not change in evolution. Upon single mutation, the free energy change $\Delta G_{\mathrm{mut}} = \Delta G_{\mathrm{wt}} + \Delta\Delta G$ is predicted adopting some models of protein stability (see below).

### Equilibrium distribution

Another approximation that is often used in these models is that the mutation rate is extremely slow ($N\mu \ll 1$) so that at every time there is only one mutant gene that "competes" with the wild-type gene for fixation in the population with effective population size of $N$ individuals. Under this scenario, the probability that the mutation gets fixed in the population can be computed with Kimura's formula (*Kimura, 1962*) as

$$P_{\mathrm{fix}}\left(\mathbf{A}^{\mathrm{wt}} \to \mathbf{A}^{\mathrm{mut}}\right) = \frac{e^{-\left(\varphi(\mathbf{A}^{\mathrm{mut}}) - \varphi(\mathbf{A}^{\mathrm{wt}})\right)} - 1}{e^{-N\left(\varphi(\mathbf{A}^{\mathrm{mut}}) - \varphi(\mathbf{A}^{\mathrm{wt}})\right)} - 1} \tag{2}$$

where $\varphi(\mathbf{A}) = \log(f(\mathbf{A}))$ is the logarithmic fitness associated with the amino acid sequence $\mathbf{A}$, Eq. (1). As it is well known, the fixation probability tends to the neutral limit $P_{\text{fix}} = 1/N$ when $\Delta\varphi$ tends to zero, it tends exponentially to zero when $\Delta\varphi$ is negative and large, and it tends to $1 - e^{-\Delta\varphi}$ when $\Delta\varphi$ is positive. Nearly neutral mutations with selective effect $|\Delta\varphi| \approx 1/N$ are likely to be fixed even when their effect is deleterious (*Ohta, 1976*). Importantly, the above fixation probability defines a Monte Carlo process in sequence space that fulfils detailed balance, so that its stationary distribution can be computed exactly (*Sella & Hirsh, 2005*, *Mustonen & Lässig, 2005*), except for the normalization constant, which would require a sum over $20^L$ possible sequences $\mathbf{A} = A_1 \cdots A_L$:

$$P(A_1 \cdots A_L) \propto \exp((N-1)\varphi(A_1 \cdots A_L)) \tag{3}$$

Note the analogy between this formula and the Boltzmann distribution with energy equal to $-\varphi$ and temperature equal to $1/(N-1)$. This explicit formula holds when the mutation process is unbiased, so that all sequences are equally probable under the mutation model. In the presence of mutation bias, the stationary distribution can be determined as the distribution with minimal Kullback–Leibler divergence from the mutational distribution, $d_{\text{KL}} = \sum_A P^{\text{mut}}(\mathbf{A})\left[\log(P^{\text{mut}}(\mathbf{A})) - \log(P(\mathbf{A}))\right]$, with a constraint on the average fitness $\sum_A P(\mathbf{A})\varphi(\mathbf{A})$. This condition generalizes the Boltzmann principle, and it was adopted for developing the mean-field model of protein evolution (*Arenas, Sanchez-Cobos & Bastolla, 2015*).

## Mean-field model of protein evolution

The mean-field (MF) model assumes that the equilibrium amino acid distribution is the product of independent distributions at each protein site,

$$P(A_1, \ldots, A_L) = \prod_{i=1}^{L} P^i(A_i). \tag{4}$$

Of course this assumption is not realistic, since different sites determine protein stability through their interactions, but it is needed for performing likelihood computations in an efficient way. Our strategy consists in determining the effect of a mutation at site $i$ self-consistently, with respect to the MF distribution at all other sites. For simplicity, we shall sometimes use the vectorial notation $P_a^i$ for indicating $P^i(a)$, where $a$ denotes one of the twenty amino acid types.

The mean-field distribution is determined by minimizing the Kullback–Leibler divergence (distance between distribution) with respect to a global mutational distribution $P_a^{\text{mut}}$, i.e., $\sum_{ia} P_a^i \log(P_a^i / P_a^{\text{mut}})$. We impose a constraint on the average fitness, which is transformed into a constraint on the folding free energy $\Delta G$. This condition on stability is imposed through the Lagrange multiplier $\Lambda$ that represents the strength of selection and is related with the effective population size. Furthermore, we impose the normalization constraints $\sum_a P_a^i = 1$ at all sites.

Since the parameters that determine the folding free energy are fixed for all proteins (see below), the only free parameters of the model are $\Lambda$ and $P_a^{\text{mut}}$. The frequencies are generally determined from the observed sequences in the protein of known structures

and the other sequences of the protein family, while $\Lambda$ is determined by maximizing the log-likelihood of the PDB sequence, $\sum_i \log\left(P^i(A_i^{\mathrm{PDB}})\right)$, which yields a well-defined single maximum. The pre-computation of the moments of the contacts makes the computation very fast, it runs in a few minutes even for proteins of several hundreds of amino acids. For further computational details see (*Arenas, Sanchez-Cobos & Bastolla, 2015*).

## Wild-type model of protein evolution

In the wild-type model (*Jimenez, Arenas & Bastolla, 2018*), we also assume that sites evolve independently. We further assume that the site-specific distribution $P_a^i$ of amino acid $a$ at position $i$ is proportional to the background distribution $P_a^{\mathrm{mut}}$ multiplied by the exponential of the logarithmic fitness of the corresponding mutation in which the wild-type amino acid in the PDB $A_i^{WT}$ is substituted by the new amino acid $a$:

$$P_a^{\mathrm{WT},i} \propto P_a^{\mathrm{mut}} \exp\left(\Lambda \varphi\left(\mathrm{mut}(A_i^{WT} \to a)\right)\right). \tag{5}$$

The fitness of a sequence is computed as in Eq. (1). The parameter $\Lambda$ is again determined by maximizing the likelihood of the wild-type sequence, $\sum_i \log\left(P^{\mathrm{WT},i}(A_i^{\mathrm{WT}})\right)$.

## Sequence entropy

The sequence entropy at position $i$ measures the variability of this position as

$$S_i = -\sum_{a=1}^{20} P_a^i \log(P_a^i), \tag{6}$$

where $P_a^i$ is obtained either from the evolutionary model (mean-field or wild-type) or from a MSA or from pooled amino acids at equivalent structural positions with the same number of contacts.

## Halpern-Bruno exchangeability matrices

To fully specify the site-specific substitution processes, besides the site-specific frequencies $P_a^i$ we need to compute consistent exchangeability matrices with the Halpern-Bruno formulas (*Halpern & Bruno, 1998*).

Given a site-specific amino acid distribution that reflects selective constraints, the Halpern-Bruno method allows computing the rate matrices of the associated site-specific substitution processes $Q_{ab}^i = E_{ab}^i P_b^i$ that are produced by a global (not site-specific) mutation process consistent together with Kimura's fixation probability, Eq. (2).

Without loss of generality, we parametrize the rate matrix of the global mutation process as $Q_{ab}^{\mathrm{mut}} = E_{ab}^{\mathrm{mut}} P_b^{\mathrm{mut}}$, where $P_a^{\mathrm{mut}}$ is the stationary matrix of the mutation process and $E_{ab}^{\mathrm{mut}}$ is its exchangeability matrix. To simplify formulas, here we assume detailed balance, i.e., we assume that $E_{ab}^{\mathrm{mut}}$ is a symmetric matrix (this condition can be easily relaxed). We write the rate matrices as $Q_{ab}^i = Q_{ab}^{\mathrm{mut}} P_{\mathrm{fix}}(f_a^i, f_b^i)$, where $f_a^i$ is the "fitness" of amino acid $a$ at site $i$. We impose that $P_{\mathrm{fix}}$ is the fixation probability Eq. (2). Halpern and Bruno showed that the site-specific fitness can be inferred from the stationary distribution from $P_a^i = P_a^{\mathrm{mut}}\left(f_a^i\right)^N$, yielding the following site-specific substitution process

$$Q_{ab}^i = E_{ab}^i P_b^i \tag{7}$$

$$E_{ab}^i = E_{ab}^{\mathrm{mut}} \left( \frac{\ln(F_b^{\mathrm{sel},i}) - \ln(F_a^{\mathrm{sel},i})}{F_b^{\mathrm{sel},i} - F_a^{\mathrm{sel},i}} \right) \tag{8}$$

$$\text{with } F_a^{\mathrm{sel},i} = \frac{P_a^i}{P_a^{\mathrm{mut}}} \tag{9}$$

The selective factors $F_a^{\mathrm{sel},i}$ quantify how much the site-specific distribution $P_a^i$ deviates from the background distribution $P_a^{\mathrm{mut}}$ induced by mutation alone.

It can be immediately seen that the exchangeability matrices $E_{ab}^i$ are symmetric, which implies that detailed balance holds and $P_a^i$ is the stationary distribution.

### Evolutionary rates

For neutral substitutions with $F_a^{\mathrm{sel},i} = F_b^{\mathrm{sel},i}$, in particular synonymous substitutions $a = b$, applying l'Hopital's rule we find $E_{ab}^i = E_{ab}^{\mathrm{mut}}/F_b^{\mathrm{sel},i}$ and $Q_{ab}^i = Q_{ab}^{\mathrm{mut}}$, i.e., the rate of synonymous substitutions equals the mutation rate, in agreement with Kimura's theory. If the amino acid $b$ is favored by selection with respect to amino acid $a$, $F_b^{\mathrm{sel},i} > F_a^{\mathrm{sel},i}$, then the substitution rate is enhanced with respect to the neutral rate, and it is decreased in the opposite case. Because of detailed balance, the flux in one direction and the other are equal, $R_{ab}^i = P_a^i P_b^i E_{ab}^i = R_{ba}^i$, with

$$R_{ab}^i = \left( P_a^{\mathrm{mut}} P_b^{\mathrm{mut}} E_{ab}^{\mathrm{mut}} \right) F_a^{\mathrm{sel},i} F_b^{\mathrm{sel},i} \frac{\ln(F_b^{\mathrm{sel},i}) - \ln(F_a^{\mathrm{sel},i})}{F_b^{\mathrm{sel},i} - F_a^{\mathrm{sel},i}} \tag{10}$$

In the above equation, the flux is partitioned into a global component that is attributed to the mutation process (superscript mut) and a site-specific component that is attributed to selection (superscript sel), which allows analysing the contributions of mutation and selection separately. The flux is maximal for substitutions $ab$ that have large and almost equal selective factors $F_a^{\mathrm{sel},i} \approx F_b^{\mathrm{sel},i}$ and have large mutational flux $P_a^{\mathrm{mut}} P_b^{\mathrm{mut}} E_{ab}^{\mathrm{mut}}$. The site-specific substitution rates are computed as the weighted average of the substitution rate matrix $Q_{ab} = E_{ab}^i P_b^i$,

$$R^i = \sum_{a \neq b} P_a^i E_{ab}^i P_b^i = \sum_{a \neq b} R_{ab}^i. \tag{11}$$

Since the flux between any pair of amino acids $a$ and $b$ decreases when their difference of fitness increases, *Halpern & Bruno, (1998)* argued that the substitution rate $R^i$ is higher at position with higher sequence entropy. However, this argument is not rigorously valid, since it neglects the fact that the substitution rate is enhanced at sites where the site-specific selective favtors $F_a^{\mathrm{sel},i} F_b^{\mathrm{sel},i}$ are correlated with the global mutational flux $\left( P_a^{\mathrm{mut}} P_b^{\mathrm{mut}} E_{ab}^{\mathrm{mut}} \right)$. Consistent with this argument, we observed that the substitution rate is not a strictly increasing function of sequence entropy but, for the same sequence entropy, it tends to increase at sites that favor polar amino acids (see 'Results'), which are characterized by higher mutational fluxes than hydrophobic amino acids.

### Mutation process

Finally, we have to define the global exchangeability matrix $E_{ab}^{\mathrm{mut}}$ that characterizes the mutation process. For this, we consider four types of mutational models. To compare

the resulting substitution rates, in all cases we fix the scale of the exchangeability matrix equating the substitution rate under mutation alone, $\sum_{a\neq b} P_a^{\mathrm{mut}} P_b^{\mathrm{mut}} E_{ab}^{\mathrm{mut}} = 1$.

1. In the first model, the global exchangeability matrix is equal to the empirical exchangeability matrix (WAG, *Whelan & Goldman, 2001*; or JTT, *Jones, Taylor & Thornton, 1992*), i.e., $E_{ab}^{\mathrm{mut}} \equiv E_{ab}^{\mathrm{emp}}$. We call this model the empirical (emp) exchangeability matrix. Since empirical substitution processes include information both on mutation and selection, we expect that they strongly correlate with the selection process.

2. In the second model, we remove the effect of selection from the empirical substitution model by imposing that for each pair of amino acids, the flux predicted by the global model and averaged over all positions is equal to the empirical flux $P_a^{\mathrm{emp}} P_b^{\mathrm{emp}} E_{ab}^{\mathrm{emp}}$, which is the observational data from which empirical models are deduced:

$$\left(P_a^{\mathrm{mut}} P_b^{\mathrm{mut}} E_{ab}^{\mathrm{flux}}\right) \frac{1}{L} \sum_i F_a^{\mathrm{sel},i} F_b^{\mathrm{sel},i} \frac{\ln(F_b^{\mathrm{sel},i}) - \ln(F_a^{\mathrm{sel},i})}{F_b^{\mathrm{sel},i} - F_a^{\mathrm{sel},i}} = P_a^{\mathrm{emp}} P_b^{\mathrm{emp}} E_{ab}^{\mathrm{emp}} \qquad (12)$$

where we use more compact matricial notation. We call the corresponding exchangebility matrix $E_{ab}^{\mathrm{flux}}$ the flux matrix (flux). This mutation model yields optimal results in phylogenetic inference (*Arenas, Sanchez-Cobos & Bastolla, 2015*).

3. Thirdly, we model the mutational process at the nucleotide level, using the genetic code and parameterizing the process through the nucleotide frequencies and the transition-transversion ratio $\kappa$. The four free parameters are fixed by imposing that the resulting background distribution $P_a^{\mathrm{mut}}$ yields amino acid frequencies as close as possible to those observed in the data, $P_a^{\mathrm{obs}}$ (*Arenas, Sanchez-Cobos & Bastolla, 2015*), as detailed below. We call the corresponding exchangeability matrix the optimized nucleotide (nuc_opt) matrix.

4. The last model is identical to the nuc_opt model, except that the nucleotide frequencies are not optimized but they are input parameters. In this way, we can vary the average hydrophobicity of the complete model by varying the Thymine content, since hydrophobic amino acids are enriched in the T base at second codon position. We call this model the nuc_var model.

In the nuc models, for any set of nucleotide frequencies and transition-tranversion rate we combine the substitution process at the nucleotide level with a selection process that assigns fitness one to sense codons and fitness zero to stop codons. Detailed balance is fulfilled at the nucleotide level, but it is only approximated at the codon level because of this selection against stop codons, therefore the transition to transversion rate can influence the stationary frequencies and we have to compute the stationary distribution of the 61 sense codons numerically.

More precisely, we model the mutation rate between two codons differing at one position, say the third one ($n_1 n_2 n_3$ and $n_1 n_2 n_3'$) as $\mu \kappa(n_3, n_3') f^{\mathrm{nuc}}(n_3') S(n_1 n_2 n_3')$, where $\mu$ is a global rate parameter, $\kappa(n_3, n_3')$ is one if $n_3, n_3'$ are related through a transversion and is the transition-tranversion rate otherwise, $f^{\mathrm{nuc}}(n_3')$ is the stationary frequency of the new nucleotide and $S(n_1 n_2 n_3')$ is zero if $n_1 n_2 n_3'$ is a stop codon, one otherwise. After the frequencies of the 61 sense codons evolve to their equilibrium state, the stationary frequencies of amino acids $P_a^{\mathrm{mut}}$ are computed summing over codons and
the exchangeability matrix is computed from the equilibrium fluxes between pairs of codons that code for any pair of amino acids. In the nuc_opt model, the score of each set of mutation parameters is computed as the likelihood of the observed number of amino acids, $\sum_a n^{\mathrm{obs}}(a)\log\left(P_a^{\mathrm{mut}}\right)$, and the parameters that maximize the likelihood are chosen.

## Data and observed substitution rates

We performed our computations on 213 proteins that were examined in a previous study (*Echave, Jackson & Wilke, 2015*). The results were qualitatively identical from one protein to the other.

The observed substitution rates of 213 proteins that we show for comparison were estimated in (*Echave, Jackson & Wilke, 2015*) from the MSA of homologous sequences through the program Rate4Site (*Pupko et al., 2002*), which builds the phylogenetic tree using a neighbour-joining algorithm (*Saitou & Nei, 1987*) and estimates rates with an empirical Bayesian approach adopting the JTT model of sequence evolution (*Jones, Taylor & Thornton, 1992*). The multiple sequence alignments were generously provided by Julián Echave and are publicly available at the url https://github.com/wilkelab/therm_constraints_rate_variation.

## Modelling stability against unfolded and misfolded states

Finally, for completeness we descibe here how we estimate the folding free energy $\Delta G$ of the experimentally known native state of a protein.

For this purpose, we adopt the contact matrix representation of the protein structure, consisting in the following: for each pair of residues at positions $i$ and $j$ along the polypeptidic chain, $C_{ij}$ equals one if the residues are in contact and zero otherwise. We define two residues to be in contact if any pair of their heavy atoms are closer than 4.5 Å. Since contacts with $|i-j| \le 2$ are formed in almost all structures, they do not contribute to the free energy difference between the native and the misfolded ensemble, and we set $C_{ij} = 0$ if $|i-j| \le 2$. The free energy of a protein in the mesoscopic structure described by $C_{ij}$ is modelled as a sum of contact interactions, $E(C,A) = \sum_{i<j} C_{ij} U(A_i, A_j)$, which depends on the type of amino acids in contact $A_i$ and $A_j$ and on 210 contact interaction parameters $U(a,b)$, for which we adopt the parameters determined in (*Bastolla, Vendruscolo & Knapp, 2000*).

For simplicity, we neglect the conformational entropy of the folded native state and estimate its free energy as $G_{\mathrm{nat}}(C^{\mathrm{nat}}, A) \approx \sum_{i<j} C_{ij}^{\mathrm{nat}} U(A_i, A_j)$. Regarding the unfolded state, we neglect their contact interactions and estimate its free energy as $G_U \approx -TLS_U$, where $T$ is the temperature in units in which $k_B = 1$, $L$ is chain length and $S_U$ is the conformational entropy per residue of an unfolded chain. We compute the free energy of the misfolded state from the partition function of the contact energy $E(C,A)$ over a set of compact contact matrices $C$ of $L$ residues that are obtained from the PDB. In agreement with previous studies (*Garel & Orland, 1988*; *Shakhnovich & Gutin, 1989*; *Bryngelson et al., 1995*), the resulting free energy is approximately described by the Random Energy Model (REM) (*Derrida, 1981*), with the addition of the third moment of the contact energy

(*Minning, Porto & Bastolla, 2013*):

$$G_{\mathrm{misf}} \equiv -T\log\left(\sum_C e^{-\sum_{i<j} C_{ij} U(A_i, A_j)/T + S(C)}\right)$$

$$\approx \langle E\rangle - \frac{\langle (E-\langle E\rangle)^2\rangle}{2T} + \frac{\langle (E-\langle E\rangle)^3\rangle}{6T^2} - LS_C T \tag{13}$$

where $LS_C$ is the logarithm of the number of compact contact matrices, $\langle.\rangle$ represents the average over the set of alternative compact contact matrices of $L$ residues. This estimate only holds above the freezing temperature, while the free energy is kept constant below the freezing temperature (*Derrida, 1981*). We assume for simplicity that the conformational entropy, $S(C_{ij})$, is approximately the same for all compact structures including the native one, and it can be neglected for computing free energy differences. The mean values of the energy can be computed from the mean values of the contacts, which are computed at the beginning and tabulated to accelerate the computation: $\langle E\rangle = \sum_{i<j}\langle C_{ij}\rangle U_{ij}$, $\langle (E-\langle E\rangle)\rangle = \sum_{i<j,k<l}(\langle C_{ij}C_{kl}\rangle - \langle C_{ij}\rangle\langle C_{kl}\rangle) U_{ij}U_{kl}$ with $U_{ij} = U(A_i, A_j)$. We also adopt the approximation that $\langle C_{ij}\rangle$ only depends on $|i-j|$ (*Minning, Porto & Bastolla, 2013*).

Putting together these free energy estimates, we obtain the free energy difference between the native and the non-native states as

$$\Delta G(C^{\mathrm{nat}}, A) = G_{\mathrm{nat}} - kT\log\left(e^{-G_{\mathrm{misf}}/kT} + e^{-G_U/kT}\right), \tag{14}$$

where the free energy of the non-native state is computed as a Boltzmann average, which is essentially equal to $G_{\mathrm{misf}}$ when the sequence is hydrophobic ($G_{\mathrm{misf}} - G_U/kT \ll -kT$) and is essentially equal to $G_U$ when the sequence is hydrophylic ($G_{\mathrm{misf}} - G_U/kT \gg kT$). For neglecting stability against misfolding, we compute $\Delta G = G_{\mathrm{nat}}(C^{\mathrm{nat}}, A) + LS_U$.

## RESULTS

### Dependence of the substitution rate on the global exchangeability model

In this work we studied two measures of the evolutionary variability of protein sites, sequence entropy and substitution rate, predicted through the site-specific stability constrained substitution models that we introduced and studied recently (*Arenas, Sanchez-Cobos & Bastolla, 2015*; *Jimenez, Arenas & Bastolla, 2018*). The results that we present arise from the predictions of our computational models, not from the analysis of natural protein sequences, thus they ignore important aspects of protein biology such as active sites and protein-protein interactions. We have to pay this price in order to address general questions such as the comparison between the two measures of evolutionary variability and how they are affected by the mutational process and by selection on protein stability.

The models adopted in this study are described in Methods and schematically represented in Fig. 1. They simulate an evolutionary process where mutations follow a global mutational process modelled through a global amino acid distribution $P_a^{\mathrm{mut}}$ and exchangeability matrix $E_{ab}^{\mathrm{mut}}$ (Fig. 1A), and they modify the stability of the protein $\Delta G$, which determines fitness through Eq. (1) (Fig. 1B). We adopt the approximation that sites of the protein evolve

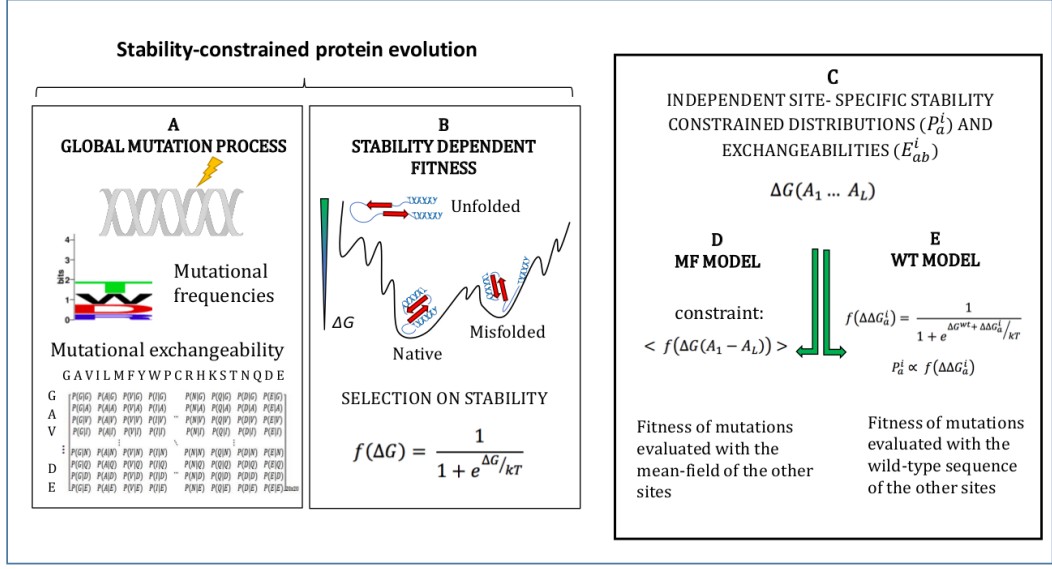

**Figure 1** Schematic representation of the site-specific stability constrained substitution models studied in this work.

independently under site-specific substitution processes globally governed by the fitness function of Eq. (1) (Fig. 1C). Since the stability $\Delta G$ depends on pairwise interactions, the fitness effect of a mutation at a site depends on the amino acids present at all other sites. In order to compute independent amino acid frequencies at each site, we perform two type of approximations: the mean-field approximation (Fig. 1D; *Arenas, Sanchez-Cobos & Bastolla, 2015*), which evaluates the effect of the mutation on stability considering the mean-field of the other sites, and the wild-type approximation (Fig. 1E; *Jimenez, Arenas & Bastolla, 2018*), which evaluates the effect of the mutation on stability when the other sites are occupied by the same amino acid as in the PDB sequence. We then compute site-specific substitution processes obtained by combining the global mutational process and the site-specific fixation probabilities computed from the site-specific amino acid frequencies through the Halpern-Bruno formula, Eq. (10).

We briefly report here previous results obtained with the stability-constrained models. Since the main component of contact interaction matrices like the one that we adopt here is hydrophobicity (i.e., $U(a,b) \approx \epsilon h(a)h(b)$, where $h(a)$ is related with the hydrophobicity of amino acid $a$), the site-specific amino acid frequencies obtained under the stability-constrained model yield high average hydrophobicity at buried sites at which the native contact matrix has many contacts. More precisely, sites constrained to be highly hydrophobic have large components of the principal eigenvector of the contact matrix (*Bastolla et al., 2005*) or, almost equivalently, large effective connectivity (*Bastolla et al., 2008*). These structural descriptors are strongly related with the number of contacts but do not coincide with them. These stability constraints in turn influence the variability of the amino acid distribution (*Porto et al., 2005*) in such a way that sites that are constrained to have either very high or very low average hydrophobic (averaged over the site-specific

amino acid distribution) are characterized by low entropy, while sites that can accomodate both polar and hydrophobic amino acids are characterized by high entropy. Because of this reason, the plot of the sequence entropy versus hydrophobicity and versus number of contacts has a bell shape.

These stability constraints that influence the sequence entropy also influence the site-specific substitution rates. However, we found that the substitution rates depend not only on the selective forces that act specifically at each protein site, but also on the global exchangeability matrix that represents the mutation process.

We considered three models of global exchangeability matrices (see 'Materials and Methods'): (1) Empirical (emp) exchangeability matrices, such as the familiar WAG (*Whelan & Goldman, 2001*) or JTT (*Jones, Taylor & Thornton, 1992*) matrices; (2) Flux exchangeability matrices (flux), which are obtained from empirical exchangeability matrices removing the selective factors represented in the stability-constrained mean-field model, so that the average flux predicted by the model between any pair of amino acids coincides with the observed empirical flux, see Eq. (12); (3) Exchangeability matrices between amino acids obtained from a mutational process at the nucleotide level with parameters optimized by maximizing the likelihood of the observed amino acid composition (nuc_opt); (4) Exchangeability matrices obtained from a mutational process at the nucleotide level with varying parameters, that allows studying the effect of varying hydrophobicity (nuc_var).

We found that empirical exchangeability matrices (emp) produce the larger substitution rates (Figs. 2B and 2C ). These matrices take into account both the mutation process and the selection process, since they have been obtained from substitutions that have been fixed through natural selection. From Eq. (10) we can see that the substitution rate is enhanced when the site-specific selective factors $F_a^{\mathrm{sel},i} F_b^{\mathrm{sel},i}$ are correlated with the global mutational flux $\left(P_a^{\mathrm{mut}} P_b^{\mathrm{mut}} E_{ab}^{\mathrm{mut}}\right)$. Since the empirical substitution models were determined in such a way that their flux equals the flux observed in real data, which accounts both for the mutational process and for selection, we expect and find that the empirical flux is strongly correlated with the selective factors $F_a^{\mathrm{sel},i} F_b^{\mathrm{sel},i}$ averaged across all protein sites. This argument explains why the empirical exchangeability matrices yield the highest substitution rates.

The flux exchangeability matrices remove from the empirical exchangeability matrix the effect of natural selection that is represented in the mean-field model averaged across sites. Consistently, we find that the substitution rates determined through the flux model are smaller than those determined with the emp model (Figs. 2B and 2C). We also found in previous work that the flux model yields larger likelihood in phylogenetic inference (*Arenas, Sanchez-Cobos & Bastolla, 2015*). Because of these results, the flux model is our default exchangeability model.

Figure 2C shows that the nuc_opt model with mutations at the nucleotide level and optimized parameters produces lower substitution rates than the flux model when associated with the MF model of selective constraints, which indicates that the flux model combined with the MF model may still include some selection. However, when the WT model of selection is applied, the nuc_opt model again produces lower substitution rates than the flux model for exposed sites with few contacts and high entropy, but the flux model

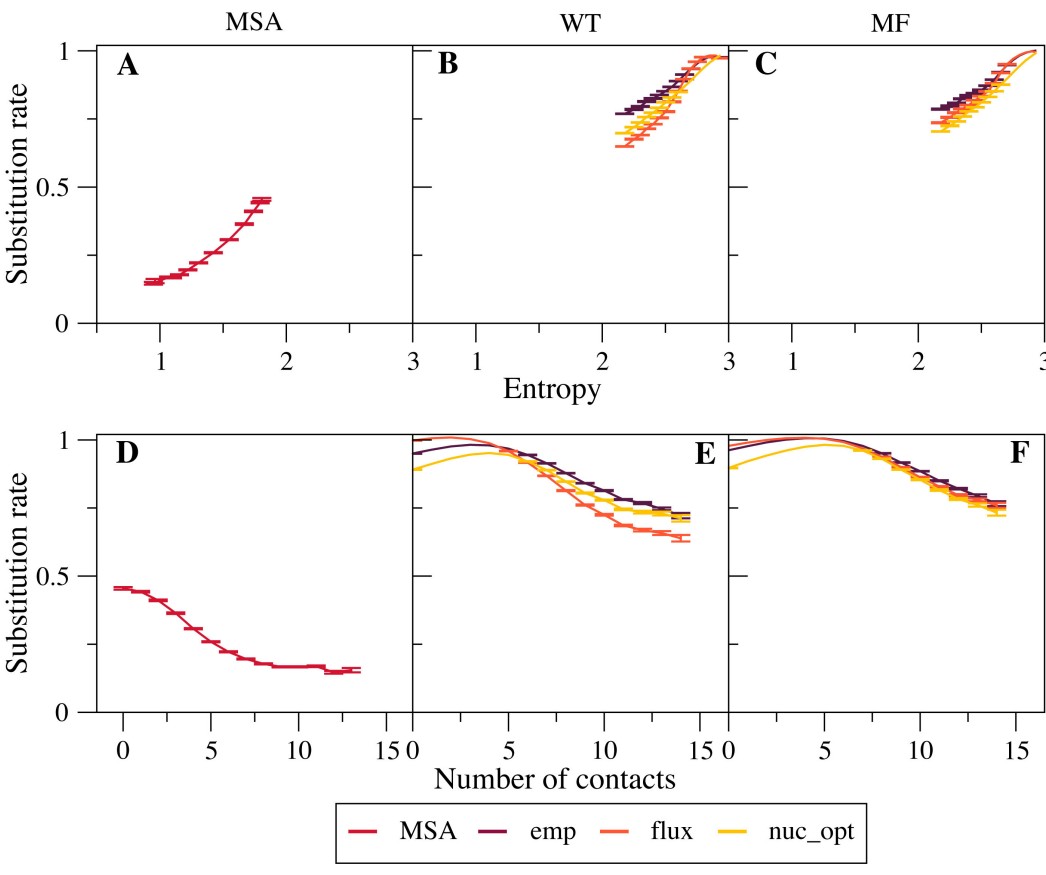

**Figure 2  Effect of the exchangeability model on substitution rates.** The plots represent substitution rate versus sequence entropy (A–C) and versus number of native contacts (D–F) for MSA (A, D), WT model (B, E) and MF model (C, F). Simulations are performed with the emp, flux and nuc_opt models of the global exchangeability matrix. In all cases the emp model produces the highest substitution rates, consistent with the fact that this model also represents selection.

produces lower substitution rates for buried sites with many contacts and low entropy (Figs. 2B and 2E). Note that the WT model represents stronger selective constraints than the MF model, since it generally predicts lower sequence entropies and substitution rates. Thus, these results suggest that the flux model associated with the WT model is effective in removing selective constraints for sites with many contacts, but less effective for sites with few contacts and high entropy.

Note that the curves that represent the substitution rates versus the sequence entropy tend to collapse for very high rates. This is due to the fact that all the global exchangeability matrices are normalized in such a way that their average flux equals one, $\sum_{ab} P_a^{\mathrm{mut}} P_b^{\mathrm{mut}} E_{ab}^{\mathrm{mut}} = 1$. This flux is achieved at neutral sites where the selective factors $F_a^{\mathrm{sel},i}$ are equal and the entropy is maximal.
## Substitution rates are different for hydrophobic and hydrophylic sites with the same entropy

Next, we investigated more in detail the relationship between site-specific sequence entropies and substitution rates. Since the flux between any pair of amino acids $a$ and $b$, Eq. (10), decreases when their difference of fitness increases, Halpern and Bruno argued that the substitution rate $R^i$ is higher at position with higher sequence entropy (*Halpern & Bruno, 1998*). However, this argument is not rigorously valid, since it neglects the fact that the substitution rate is enhanced at sites where the site-specific selective factors $F_a^{\mathrm{sel},i}F_b^{\mathrm{sel},i}$ are correlated with the global mutational flux $\left(P_a^{\mathrm{mut}}P_b^{\mathrm{mut}}E_{ab}^{\mathrm{mut}}\right)$. One can see in Figs. 2B–2C that sites with larger entropy tend to have on the average larger substitution rates, as predicted by *Halpern & Bruno (1998)*, but the substitution rate is not a strictly increasing function of sequence entropy, not even when it is averaged over different sites.

We then show in Fig. 3 the detailed plot of the substitution rate versus the sequence entropy for all sites of a small protein, chosen in such a way that we can spot all of the sites. We can clearly see in Fig. 3 two branches that correspond to different numbers of native contacts. Sites with few contacts, which tend to be occupied by polar amino acids, evolve faster than sites with many contacts, occupied by hydrophobic amino acids, even if their sequence entropy is equal. With the flux model of the mutation process, which we consider the most reliable model since it reproduces the empirically observed flux between all pairs of amino acids, this happens for both the MF and the WT model of natural selection (plots C and D). All other studied proteins present the same trend (see Figs. S1 and S2), but for large proteins the representation is less clear. When we model the mutation process at the codon level through the model nuc_opt, whose parameters are separately optimized for each protein, the differences between the two branches decrease considerably, in particular when we apply the WT model of selection (Fig. 3B) and the trend may change from one protein to the other, since different proteins evolve under different mutation processes, in such a way that the buried branch may evolve faster than the exposed branch for some protein, see Figs. S3 and S4.

Since sequence entropy is a measure of the selective constraints, this difference between sites with equal sequence entropy must be attributed to the mutation process embodied in the exchangeability matrix, not to natural selection. The explanation of this finding is based on Eq. (10): polar amino acids tend to have higher mutational fluxes $\left(P_a^{\mathrm{mut}}P_b^{\mathrm{mut}}E_{ab}^{\mathrm{mut}}\right)$ than hydrophobic amino acids, therefore exposed sites in which polar amino acids have larger site-specific selective factors $F_a^{\mathrm{sel},i}F_b^{\mathrm{sel},i}$ tend to evolve faster than buried sites.

This result contradicts the expectation that the site-specifi substitution rates are monotonic functions of the sequence entropy (*Halpern & Bruno, 1998*): one can see from Fig. 3 that sites characterized by lower entropy can substitute faster if they are exposed sites occupied by polar amino acids.

One may note that in Fig. 3 some white points with small number of contacts overlap with the black points with large number of contacts. This is due to the fact that the number of contacts is only an approximate predictor of the selective factors favoring hydrophobicity, while better correlation with the average hydrophobicity is achieved using the principal eigenvector of the contact matrix (*Bastolla et al., 2005*) or the effective connectivity profile

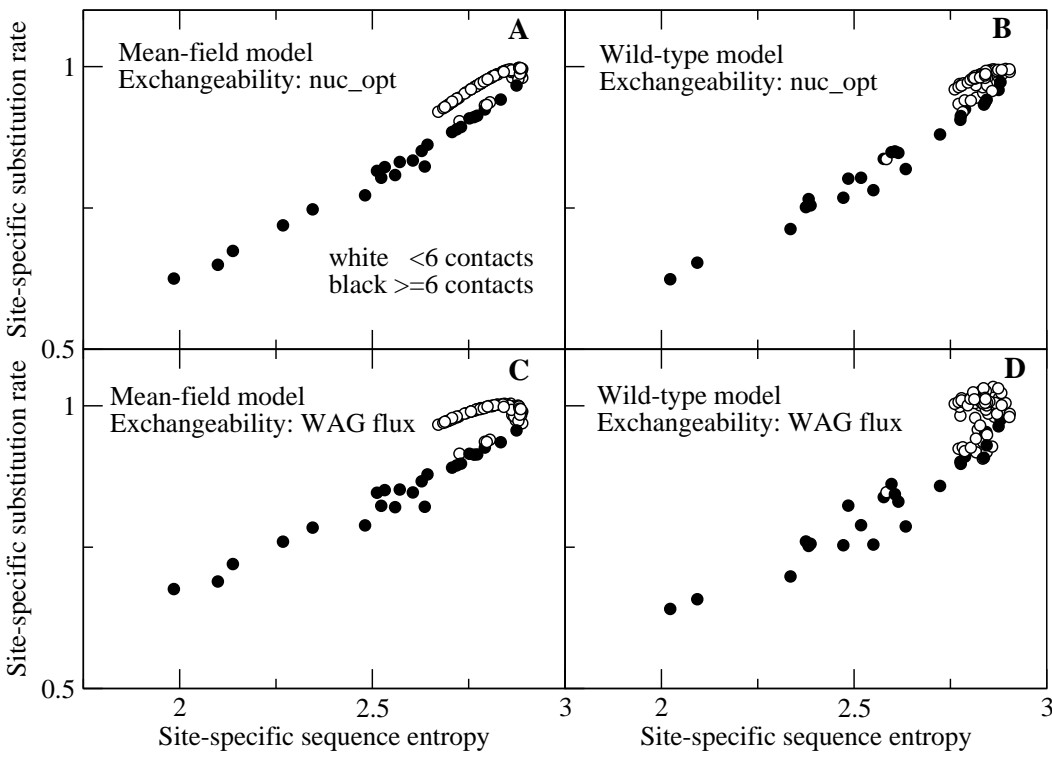

**Figure 3** **Each point represents the substitution rate versus the sequence entropy for all sites of the ribonuclease protein with PDB code 1pyl, which is representative of our data set and makes the figure easier to interpret because of its small number of sites.** One can spot two branches, corresponding to sites that evolve faster and slower for the same sequence entropy. The two branches correspond to polar sites with few contacts (white circles) and hydrophobic sites with many contacts (black circles), respectively. The four plots represent various combinations of selection (MF, WT) and mutation (nuc_opt, flux) models. (A) MF and nuc_opt. (B) WT and nuc_opt. (C) MF and flux. (D) WT and flux.

that generalizes it for multidomain proteins (*Bastolla et al., 2008*), which are not merely local descriptors as the number of contacts but also represent the global topology of the native contact matrix.

## More hydrophobic proteins substitute more slowly, but mutation bias towards hydrophobicity increases the substitution rates

After investigating the relationship between hydrophobicity and substitution rates comparing individual sites, we perform the same analysis comparing different proteins. For this purpose, we group the 213 proteins in our data set according to their predicted average hydrophobicity under the same mutational process and compare the substitution rates of groups characterized by different hydophobicity. In Fig. 4, each point represents a group of proteins with similar mean hydrophobicity. Each curve is obtained for a different mutation process with its background distribution $P^{mut}$ and exchangeability matrix $E^{mut}$. One can see that, for the flux mutation process (black circles in Fig. 4), more polar proteins tend to evolve more rapidly, consistent with what we observed for polar sites in Fig. 3.

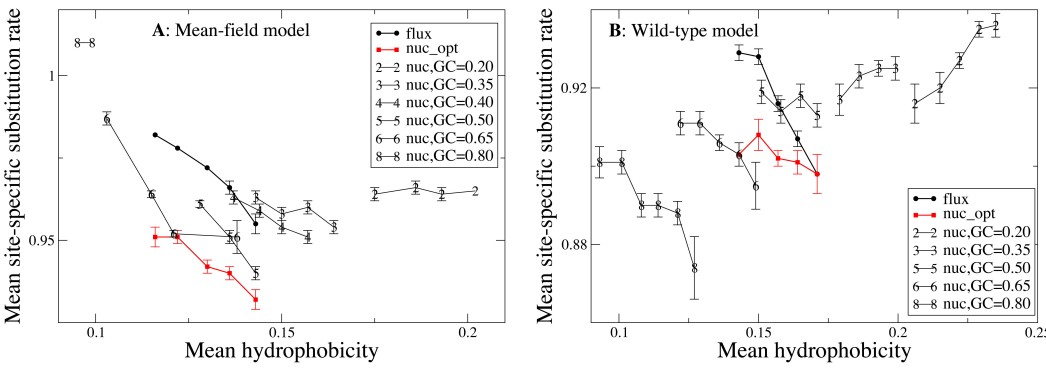

**Figure 4** **In this plot each point represents a group of proteins with similar mean hydrophobicity in the evolutionary model, and each curve is obtained by varying the global mutational distribution and exchangeability matrix, which represent the mutation process.** One can see that, for the same mutation process, more hydrophobic proteins tend to evolve more slowly, except when the mutation process induces very high hydrophobicity, in which case the substitution rate becomes an increasing function of hydrophobicity. On the other hand, mutation processes with extreme properties (very high or very low hydrophobicity) tend to increase the substitution rate. The two plots represent the two selection models. (A) MF. (B) WT. Each curve in the plots represent one mutation model: flux, nuc_opt, and five flavours of the nuc_var model with different values of the equilibrium GC content.

Once again, this behavior may be explained by considering that more polar amino acids have higher mutational fluxes.

The above also holds for the nuc_opt model, in which the mutation process is separately optimized for each protein, but in this case when the WT selection model is used the maximum of the substitution rate is achieved for proteins of intermediate hydrophobicity (see red squares in Fig. 4B), consistent with the observation that with the nuc_opt model the hydrophobic sites may evolve slightly faster than the polar sites in some proteins characterized by a mutation process that favors hydrophobic residues, see Figs. S3 and S4).

We then consider the same mutation process for all proteins, parameterized by the G+C content at the mutational equilibrium (nuc_var model). Since Thymine at second codon position almost always codes for hydrophobic amino acids, there is a negative correlation between G+C content of the mutation model and the average hydrophobicity of the protein sequence. Varying the mutation bias we construct different sets of model proteins that present varying hydrophobicity and are characterized by different mutational fluxes. In this way, we can investigate how the mutation bias influence the biophysical properties (hydrophobicity) and the evolutionary properties (substitution rate, sequence entropy) of an evolving protein. When the GC content is high and the hydrophobicity is low (GC $\geq$ 0.35 with the MF model, Fig. 4A and GC $\geq$ 0.5 with the WT model, Fig. 4B) more polar proteins tend to evolve faster, as we observed with the flux model. However, when the mutation process induces high hydrophobicity (GC = 0.2 in Fig. 4A and GC <0.5 in Fig. 4B), the substitution rate becomes an increasing function of hydrophobicity. This is easily rationalized by the fact that, when the background distribution $P_a^{\text{mut}}$ is biased towards hydrophobic amino acids, the mutational flux $P_a^{\text{mut}}P_b^{\text{mut}}E_{ab}^{\text{mut}}$ is higher between pairs of hydrophobic residues, and the other way round.

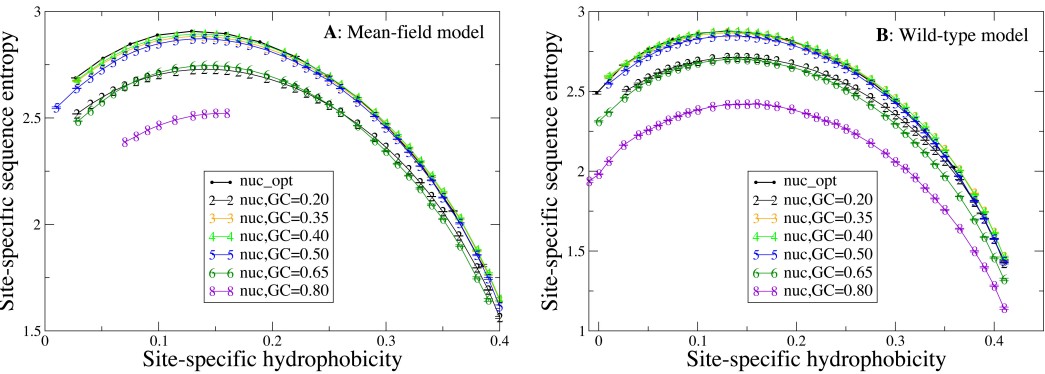

**Figure 5** **Each point represents a set of protein sites with similar average hydrophobicity in the evolutionary model.** The sequence entropy has a universal shape as a function of the average hydrophobicity, with a maximum at 0.14, which is the mean hydrophobicity of the uniform distribution of amino acids. Changes in the background distribution mostly shift the sequence entropy curves without changing the position of the maximum, but they affect the values of entropy. The largest entropies are obtained for the mutation model optimized for each protein sequence (thick black line) and for the mutation bias with GC content equal to 0.40, which yield only slightly hydrophobic sequences. The two plots represent the two selection models. (A) MF. (B) WT. Each curve in the plots represent one mutation model: nuc_opt, and five flavours of the nuc_var model with different values of the equilibrium GC content. The mutation model flux gives in this case exactly the same curve as nuc_opt and it is not represented.

Different G+C content in Fig. 4 represent different mutational processes, which may be interpreted as bacterial species characterized by different GC bias. For the MF model (plot A), one can see that mutational processes with extreme bias (very high or very low G+C content and hydrophobicity) tend to increase the average substitution rate, while for the WT model (plot B) mutational processes biased towards hydrophobic residues such as GC = 0.2 have higher substitution rates. Therefore, although hydrophobicity is negatively correlated with the substitution rate when we compare different proteins evolving with the flux mutational process (black points in Figs. 4 and Figs. 3C and 3D) or with mutational processes with high G+C, the correlation becomes positive if we compare different mutational processes.

## Influence of the mutation bias on sequence entropies

Next, we study how the shape of the entropy-hydrophobicity curve depends on the mutation bias. In Fig. 5 each point represents a set of protein sites with similar hydrophobicity in the stability-constrained evolutionary model. The sequence entropy has an almost universal bell shape as a function of the average hydrophobicity of the site, with a maximum when the average hydrophobicity is approximately 0.14, which is the average hydrophobicity of the uniform distribution of amino acids. This result is of course not surprising, since sites with very high or low average hydrophobicity have distributions that favor only the most hydrophobic or polar amino acids, while all amino acids are allowed when the average hydrophobicity equals the one of the uniform distribution.

Changes in the mutation bias do not change the position of the maximum, but they strongly shift the sequence entropy curves in the vertical direction. The largest entropies are

obtained for the mutation model nuc_opt optimized separately from each PDB sequence (thick black line) and for the mutation bias with G+C content equal to 0.40, which has a small bias towards slightly hydrophobic sequences. Extreme mutation bias both towards hydrophobic (low GC) and polar (high GC) amino acids yield very reduced sequence entropies, which means that selection must impose stronger constraints in order to preserve the average hydrophobicity needed for stable proteins. This result is consistent with the finding that, for equal population size, the average fitness achieved in evolution has a maximum as a function of the mutation bias, and it is low for extreme mutation bias either toward hydrophobic or towards hydrophylic sequences (*Mendez et al., 2010*; *Bastolla, Dehouck & Echave, 2017*).

### Influence of hydrophobicity on substitution rates

We now study the relationship between site-specific hydrophobicity and site-specific substitution rate. As in the previous figure, also in Fig. 6 each point represents a set of protein sites with similar average hydrophobicity in the stability-constrained evolutionary model, and we plot the average substitution rate versus the average hydrophobicity. Different from the shape of the sequence entropy, the shape of the substitution rate curve clearly depends on the mutation bias. The average hydropobicity at which the maximum substitution rate is achieved decreases with the G+C content or, equivalently, it increases with the average hydrophobicity of the mutation process. In other words, when the mutation process favors the exchange of hydrophobic amino acids, the maximum of the substitution rate is achieved at sites that are more hydrophobic, as expected on the basis of Eq. (10) that suggests that sites where the site-specific selective factors are correlated with the mutational flux have higher substitution rates. This result confirms that the mutation process has a strong influence on the substitution rates.

Consistent with Fig. 4, the substitution rate at the maximum tends to increase for mutation processes that favor higher hydrophobicity (lower G+C bias), but for the MF model (plot A) they also increase for extremely polar mutation bias (G+C content 0.8). Consistent with the results reported in Fig. 2, the flux model of the exchangeability matrix (thick black line) predicts higher substitution rates than the nuc_opt model (red) when applied together with the MF model (Fig. 4A), but when it is applied together with the WT model it predicts lower substitution rates at hydrophobic sites with many contacts (Fig. 4B), suggesting that the WT model is effective at removing the effect of selection from the flux model at these buried sites.

## DISCUSSION AND CONCLUSIONS

Here we studied how the evolutionary variability of proteins is influenced by the underlying mutation process, adopting a model of stability-constrained protein evolution with selection on the stability of the native state against both unfolding and misfolding.

We found that the sequence entropy and the substitution rate are not equivalent measures of the evolutionary variability of the protein sites, as it was expected based on the arguments presented in the seminal paper by *Halpern & Bruno (1998)*. These two measures are positively correlated, as seen from Fig. 2, which shows that the substitution

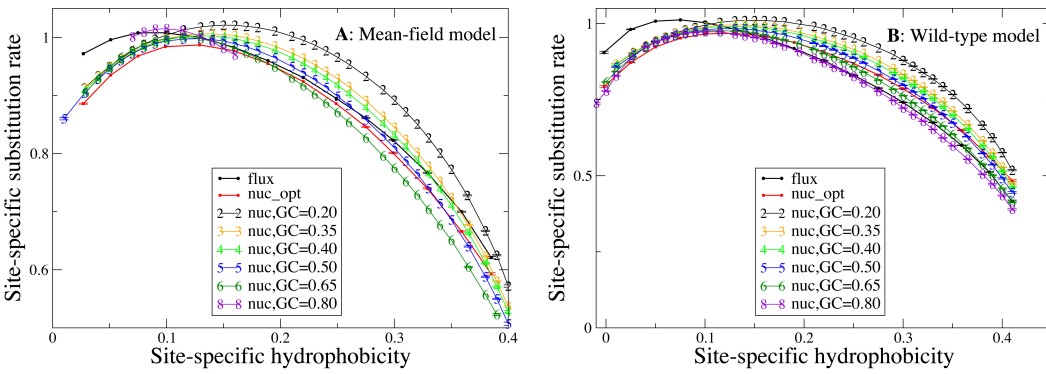

**Figure 6 Each point represents a set of protein sites with similar hydrophobicity in the evolutionary model.** The substitution rate shows a maximum whose position depends on the mutation process. The hydropobicity at which the maximum rate is achieved increases with the mean hydrophobicity of the mutation process (lower GC content). The substitution rates tend to increase for mutation processes that yield higher hydrophobicity (lower GC content), but for the MF model (plot A) they also increase for extremely polar mutation bias (GC content 0.8). The two plots represent the two selection models. (A) MF. (B) WT. Each curve in the plots represent one mutation model: flux, nuc_opt, and five flavours of the nuc_var model with different values of the equilibrium GC content.

rate tends to increase for sites with higher sequence entropy. Nevertheless, sites with the same sequence entropy are characterized by different substitution rates, which are systematically higher for polar sites than for hydrophobic sites (Fig. 3). This difference is not due to different selective constraints, which are quantified by sequence entropy, but it is due to the different exchangeability of polar and hydrophobic amino acids, which does not influence the sequence entropy but influences the substitution rate according to Eq. (10). This equation shows that the substitution rate is larger at sites where the site-specific selective factors are correlated with the global mutational flux. In particular, at exposed sites polar residues have higher selective factors, and under the flux model these residues are characterized by high mutational fluxes, which explains why exposed sites tend to evolve faster than buried sites with the same sequence entropy. This result is independent of the protein structure and robust with respect to changes of the selection model (WT or MF). As a consequence, more polar proteins are predicted to evolve faster than proteins with large mean hydrophobicity (Fig. 4).

When we apply the nuc_opt codon mutation model based on nucleotide frequencies separately optimized for each protein, we still observe that sites with the same entropy evolve with different rates depending on whether they are exposed or buried, but the differences decrease (Fig. 3) and, for some proteins, buried sites may evolve faster than polar sites with the same entropy (Figs. S3 and S4), although there is still a negative correlation between the substitution rate of a protein and its hydrophobicity (Fig. 4). This is consistent with the observation that, for low G+C mutation bias that favor hydrophobic residues, the correlation between the substitution rate and the hydrophobicity of proteins becomes positive (see Fig. 4), as expected based on Eq. (10).

We then compare different mutation biases applied to all proteins of our data set. The average substitution rates tend to be larger for mutation bias favoring hydrophobic residues (low G+C) (Figs. 4 and 6). Thus, the comparison of proteins with different hydrophobicity under the same mutation model and the comparison between different mutation processes, such as those happening in different bacterial genomes, yield contrasting results for the substitution rates: substitution rates tend to be higher for more polar proteins evolving under the same mutation process, but they tend to be higher in organisms with mutation bias towards A+T that favor hydrophobic residues, such as intracellular bacteria. Note that the substitution rates also increase for mutation bias favoring very polar amino acids (high G+C), but the latter happens only when the MF model of selection is applied.

As Eq. (10) shows, the higher substitution rate for equal sequence entropy observed at exposed sites is attributable to the higher exchangeability of polar residues, which is a property of the mutational process. In contrast, the differences in substitution rates that we observe for proteins evolving under different mutational processes (Fig. 6) is likely to be caused at least in part by natural selection. In fact, buried sites are characterized by lower entropy than exposed sites (Fig. 5), which indicates that they experience stronger selective constraints and their selective factors $F_a^{\text{sel},i}$ are more skewed towards hydrophobic residues. When the mutation bias towards A+T increases, the mutational flux $P_a^{\text{mut}} P_b^{\text{mut}} E_{ab}^{\text{mut}}$ of hydrophobic residues increases and the site-specific substitution rate given by Eq. (10) increases at buried sites characterized by skewed $F_a^{\text{sel},i}$ more rapidly than it decreases at exposed sites, as one can see from Fig. 6 that shows a large increase of the substitution rate at buried sites with large average hydrophobicity while the substitution rate at exposed sites depends little on the mutational bias.

Therefore, the substitution rate is systematically influenced by the mutation bias (Fig. 6), in such a way that when the mutation bias favors more hydophobic proteins (low G+C) the substitution rate increases and its maximum is achieved at sites that are more hydrophobic, as expected from Eq. (10)). On the contrary, the curve of the sequence entropy versus the average hydrophobicity has a shape that does not depend on the mutation bias. In particular, the position of the maximum always coincides with the value 0.14, which is the average hydrophobicity of the uniform distribution of amino acids. As a consequence, the site-specific average hydrophobicity at which the sequence entropy is maximal does not coincide in general with the average hydrophobicity at which the substitution rate is maximal, and this discrepancy becomes larger when the mutation bias is more extreme, causing larger differences between the two measures of evolutionary variability.

Finally, changes of mutation bias severely affect the selective constraints imposed on the protein sites, attaining maximum values of the entropy when the mutation bias is $G+C = 0.40$ and decreasing the site-specific sequence entropies when the mutation bias becomes more extreme both towards hydrophobic and towards polar residues (Fig. 5). The sequence entropy of exposed polar sites decreases with the mutation bias more strongly than the entropy of buried hydrophobic sites, where the different curves in Fig. 5 tend to collapse. Thus, exposed sites are affected by weaker selective constraints than buried sites (they have higher entropy, see Fig. 5), but these selective constraints become more severe when the mutation bias becomes extreme.

## ACKNOWLEDGEMENTS

We thank Julián Echave for providing us the empirical multiple sequence alignments that were used for the leftmost plots in Fig. 2, and we acknowledge interesting discussions with him, Alberto Pascual-García and Nick Goldman. Research at CBMSO is facilitated by the Fundación Ramón Areces.

### Funding

This work was supported by the Spanish Ministry of Economy through grants BIO2016-79043 and BFU2012-40020. Miguel Arenas was supported by the Ramón y Cajal Grant RYC-2015-18241 from the Spanish Government. The funders had no role in study design, data collection and analysis, decision to publish, or preparation of the manuscript.

### Grant Disclosures

The following grant information was disclosed by the authors:
Spanish Ministry of Economy: BIO2016-79043, BFU2012-40020.
Ramón y Cajal: RYC-2015-18241.

### Competing Interests

Ugo Bastolla is an Academic Editor for PeerJ.

### Author Contributions

- María José Jiménez-Santos performed the experiments, analyzed the data, prepared figures and/or tables, authored or reviewed drafts of the paper, approved the final draft.
- Miguel Arenas analyzed the data, authored or reviewed drafts of the paper, approved the final draft.
- Ugo Bastolla conceived and designed the experiments, performed the experiments, analyzed the data, contributed reagents/materials/analysis tools, prepared figures and/or tables, authored or reviewed drafts of the paper, approved the final draft.

### Data Availability

The program Prot_evol is freely available for download at https://ub.cbm.uam.es/prot_fold_evol/prot_fold_evol_soft_main.php#Prot_Evol.

The multiple sequence alignments from which the leftmost plots of Fig. 1 have been obtained are available at https://github.com/wilkelab/therm_constraints_rate_variation/.

### Supplemental Information

Supplemental information for this article can be found online at http://dx.doi.org/10.7717/peerj.5549#supplemental-information.

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
