# Peer review of "Influence of mutation bias and hydrophobicity on the substitution rates and sequence entropies of protein evolution"

_PeerJ, doi:10.7717/peerj.5549_

## Round 0.1 · original submission · Major Revisions

Both reviewers find your work interesting and novel. While reviewer 1 has a few minor comments, reviewer 2 raises more substantial points. I expect that some rewriting and some additional analysis will be required to address all the reviewer comments.

Reviewer 1 ·

Basic reporting

The English was clear and easy to read. I only found minor changes to be made:
1. Page 17 line 14: "The result robust with respect" might be "The result *is* robust with respect"
2. Page 17 line 31: "likely caused both both by" has an extra "both" in it.

I found the background to be sufficient and the references clearly cited.

The structure of the manuscript including the figures was professional and I could find the raw data at the github repository. The figures are sufficient, but I found a few small improvements:

1. Figure 1: The Entropy x axis major tick labels overlap between the WT and MF subfigures. It looks like 1 and 3 are being printed on top of each other. I believe the correct fix is to make the MF x-axis range (currently 1 to 3) the same as the MSA and WT, which appears to be 0.5 to 3.

2. Figure 1: I noticed a light blue line between the graph legend and the figure caption. It is there on 3 of my PDF viewers, but it is not in rate_entr_v3.eps, so I am not sure if it is a rendering problem or perhaps something in the LaTeX file.

3. Figures 3-5: The font size used in these figures is much smaller than the font in previous figures and the rest of the paper. I suggest increasing it for readability.

The article seems like a reasonable unit of publication and is completely self-contained.

Experimental design

The article fits in the Aims and Scope of PeerJ.


The research question is well defined, however I am not convinced the knowledge gap it fills. The primary question addressed is "whether the sequence entropy and the substitution rate are equivalent measures of the evolutionary variability of a position". The authors then show that they are not equivalent. Was there confusion in the field of their equivalence? Did a previous group publish evidence that they may be identical? Perhaps the focus could be shifted slightly to say something new about sequence entropy, substitution rates, hydrophobicity, and mutation bias.


The research seems to be done with a high technical standard using up to date models and software.

The methods are clearly and completely described. I appreciate how all of the terms and models are thoroughly defined.

Validity of the findings

The data and sources seem robust and well documented.

The conclusions followed clearly and directly from the results, in order to answer the question at hand. I found very little speculation in the conclusions.

Additional comments

The article is well structured and clearly laid out: the data are solid, the question is clear, the figures support the results which answer the question. As I stated in section 2, my main concern is the relevance of the question. Three of your five figures focus on hydrophobicity's influence on the sequence entropy and the substitution rate. A few of your conclusions describe the effects of mutational bias. Perhaps restructuring your question to include hydrophobicity and mutational bias would more clearly describe the knowledge gap that your article fills.

Reviewer 2 ·

Basic reporting

Developing a mechanistic model of protein evolution is crucial in creating a null model for the observed patterns of genetic variation among extant sequences. In particular, the evolutionary variability of an amino acid site in protein families is an important indicator of the selective constraints experienced by the site in the course of evolution. However, the observed substitution variability is a function both of mutational input (or bias if it exists) and selection. Decoupling the relative contribution of these two is an important and challenging problem in protein evolution.

In this work, Jiménez-Santos et al. study the site-specific substitution rates of proteins under selection for protein stability and misfolding. They demonstrated that substitution rates vary depending on the exchangeability of mutations, which in turn depends on the hydrophobicity of residues. They found that, on average, polar sites evolve faster unless the sequence is evolved under mutational bias that prefers hydrophobic residues. Importantly, they link the mutational bias to the biophysical properties and per-site evolutionary rates. Overall, I think the paper has merits and provides a relevant connection between biophysical properties of proteins and the observed rates of molecular evolution. However, I have several technical and conceptual points that need to be addressed prior to publication.

Experimental design

See comments below.

Validity of the findings

See comments below.

Additional comments

Major comments:
1. The authors claim, “more hydrophobic proteins substitute more slowly, but mutation bias towards hydrophobicity increases the substitution rates". But an alternative explanation could simply be selection for PPI. Proteins that have more interaction partners have higher hydrophobic surface patches, thus are enriched in their total number of hydrophobic residues. Moreover, the interaction sites are also under stronger selective constraints and will evolve slowly. Thus, the claimed trend might be due to PPI and not due to mutational bias. How many of the 230+ proteins analyzed are involved in PPI and what are their centrality in the PPI networks? See for example the following works: Qian, et al PNAS 108(21):8725-30; Heo et al. PNAS 108 (10) 4258-4263; Zhang Nature Reviews Genetics 16, 409-429 (2015).

2. In the introduction, the authors make the distinction between two kinds of models/approaches that rationalize why sites that form many contacts are subject to stronger selective constraints. The first kind is the “stability-constrained fitness models” that assume a fitness function f=1/(1+expDG/kT) and that estimates the change in free energy upon mutation “assuming that the native structure does not change”. The second model is the “structurally constrained model”, which in contrast to the first, include the effects of how random mutations might perturb the structure of the native state using the GNM.

However, I think this is a false dichotomy. Several studies that model the fitness landscape as f~1/(1+exp(DG/kT)) also account for the structural changes in the native structure because they use a physical force field and the protein 3D structure to estimate the mutational effects on folding stability DDG (e.g. Jackson & Wilke PeerJ 2013 Nov 12;1:e211; Serohijos & Shakhnovich Mol Biol Evol. 2014 31(1):165-76). Moreover, in other approaches where the fitness changes are derived from experimental data (Bloom Mol Biol Evol. 2014 Aug;31(8):1956-78; Mol Biol Evol. 2014 Oct;31(10):2753-69) the fitness effects already account for selection for folding stability and effects of mutations on structure.

3. For Equation 10, its derivative with respect to the change in fitness is zero when Fa=Fb as pointed out by the authors. This implies that not only selected mutations but also neutral mutations would contribute to a higher substitution rate. This seems in contrast with the conclusion that the empirical exchangeability produces the highest substitution rate because it contains selection. I think a clarification of how neutral mutations might clarify this point to the readers.

4. Since the authors adopt a solid mathematical approach in this paper, I strongly advise that they also work out the analytical relation between sequence entropy and substitution rates. They can take the logarithm of Eq. 11 and re-factor sequence entropy. The evolutionary rate should be exponentially dependent on sequence entropy, which seems to be the behavior depicted in Figure 1.

5. One interesting observation in Figure 1 is that the difference between different exchangeabilities diminishes at higher entropies (i.e., all curves coincide). This suggests that at low sequence entropies either (i) the natural sequences tend to have more neutral substitutions or (ii) more mutations are under selection. Perhaps the authors can elaborate on this point.

6. I might be wrong but I think the main determinant of the shape of site-specific sequence entropy versus hydrophobicity is the equilibrium probability of amino acids rather the hydrophobicity. At higher/lower per-site hydrophobicity we have more/ fewer mutations fixed. Therefore, the equilibrium distribution is shifted towards the extreme (i.e., one mutation with P=1 and hence entropy=0). I would like to see the same plot with varying charge or hydrophilicity.

7. With regards to the previous point, don’t the authors maximize the likelihood of the WT sequence as explained in Eq. 5, which already contains the background distribution Pa? If so, it should not be surprising that the maximum substitution rate, when the background is more hydrophobic, occur in sites that are more hydrophobic?
8. There are several instances where claims are made, but no data is referred:
*“The WT models show even better performance on several data sets (Arenas & Bastolla, In preparation).” (Page 3, line 32)
* “Thus, in theory, the WT model is more suited for short evolutionary divergences and MF model is more suited for long evolutionary divergences (Arenas & Bastolla, In preparation).” (Page 4, line 2)
In my opinion, these are very important and relevant claims, which require that either the data be included as supplementary figures or as an accompanying manuscript.
11) It is an important observation in their work that substitution rate Ri and sequence entropy do not correlate, which is in contrast to the more intuitive claim by (Halpern and Bruno 1998). Can the authors provide an explanation on why might this be the case?
12) Interestingly, the data in Fig 2 is only for one protein (PDB code 1PYL). Although, I do not challenge the claim that this is representative of “all of our data set.” I think the figure could be made stronger by showing for all 200+ proteins.

Minor points:
1) The manuscript can benefit if authors add a schema of the methods/models as Figure 1.
2) Also, the discussion is solely focused on the immediate consequences of the results (e.g., the effect of mutational bias on the substitution rate). As a matter of fact, sequence entropy is frequently used to detect selection in the evolution of influenza proteins such as haemagglutinin. I suggest authors comment on the applicability of their methods in a broader context.
3) In Figure 2, there is an overlap of white circles (<6 contacts) and black circles (6>= contacts) suggesting that some sites, despite having the same sequence entropy and less than 6 contacts evolve with a lower rate. Are these active site residues? It would be interesting to comment on these sites.
4) I could not access the GitHub page.

---

## Round 0.2 · accepted · Accept

Thank you for carefully addressing the reviewer comments.

# Reviewer 2 ·

Basic reporting

See below.

Experimental design

See below.

Validity of the findings

See below.

Additional comments

I think Bastolla and co-workers have already addressed the points I raised.